

# Patterns and controls of soil respiration and its temperature sensitivity in grassland ecosystems across China

Jiguang Feng[1], Jingsheng Wang[2], Yanjun Song[3], Biao Zhu[1]

[1]Department of Ecology, College of Urban and Environmental Sciences, and Key Laboratory for Earth Surface Processes of the Ministry of Education, Peking University, Beijing 100871, China

[2]Qianyanzhou Ecological Research Station, Key Laboratory of Ecosystem Network Observation and Modeling, Institute of Geographic Sciences and Natural Resources Research, Chinese Academy of Sciences, Beijing 100101, China

[3]Forest Ecology and Forest Management Group, Wageningen University and Research Centre, PO Box 47, 6700 AA Wageningen, the Netherlands

*Correspondence to:* B. Zhu (biaozhu@pku.edu.cn)

**Abstract.** Soil respiration (Rs), a key process in the terrestrial carbon cycle, is very sensitive to climate change. In this study, we synthesized 54 measurements of annual Rs and 171 estimates of $Q_{10}$ value (the temperature sensitivity of soil respiration) in grasslands across China. We quantitatively analyzed their spatial patterns and controlling factors in five grassland types, including temperate typical steppe, temperate meadow steppe, temperate desert steppe, alpine grassland, and warm-tropical grassland. Results showed that the mean ($\pm$ SE) annual Rs was $582.0 \pm 57.9$ g C m$^{-2}$ yr$^{-1}$ across Chinese grasslands. Annual Rs significantly differed among grassland types, and positively correlated with mean annual temperature, mean annual precipitation, soil organic carbon content and aboveground biomass, but negatively correlated with latitude and soil pH ($P <$ 0.05). Among these factors, mean annual precipitation was the primary factor controlling the spatial variation of annual Rs in Chinese grasslands. The mean contributions of growing season Rs and heterotrophic respiration to annual Rs were 78.7% and 72.8%, respectively. Moreover, the mean ($\pm$ SE) of $Q_{10}$ across Chinese grasslands was $2.60 \pm 0.08$,




ranging from 1.03 to 8.13, and varied largely within and among grassland types, and
among soil temperature measurement depths. Generally, the seasonal variation of soil
respiration in Chinese grasslands cannot be well explained by soil temperature using the
van't Hoff equation. Longitude and altitude were the dominant driving factors and
accounted for 26.0% of the variation in $Q_{10}$ derived by soil temperature at the depth of 5
cm. Overall, our findings advance our understanding of the spatial variation and
environmental control of soil respiration and $Q_{10}$ across Chinese grasslands, and also
improve our ability to predict soil carbon efflux under climate change on the regional
scale.
**Keywords:** Soil respiration, Soil carbon emission; Temperature sensitivity; Grassland
ecosystem; $Q_{10}$

## 1 Introduction

Soil respiration (Rs) represents carbon dioxide ($CO_2$) efflux from the soil surface,
including autotrophic/root respiration, and heterotrophic/microbial respiration associated
with soil organic matter and litter decomposition (Boone et al., 1998; Kuzyakov, 2006;
Schindlbacher et al., 2009). As one of the largest fluxes in the global carbon cycle, Rs
plays an important role in regulating ecosystem carbon cycling, carbon-climate feedback,
and climate change (Raich and Schlesinger, 1992; Davidson et al., 2002; Luo and Zhou,
2006; Bond-Lamberty and Thomson, 2010). The temperature sensitivity of Rs ($Q_{10}$), the
factor by which Rs is multiplied when temperature increases by 10 ℃, is a key parameter
to evaluate the feedback intensity between soil carbon efflux and climate warming
(Reichstein et al., 2005; Davidson and Janssens, 2006). Knowledge on patterns and
controls of Rs and $Q_{10}$ variation on the large scale is crucial for better understanding and
modeling soil carbon cycle in a warmer world (Peng et al., 2009; Wang et al., 2010).

Temperature and precipitation are commonly believed to be the most important climatic
factors controlling Rs on the large scale, as suggested by a number of studies (Raich and
Schlesinger, 1992; Raich and Potter, 1995; Chen et al., 2014; Hursh et al., 2017). As the
indirect factors, altitude and latitude can also affect Rs by affecting climatic factors (Song
et al., 2014). Soil properties, such as soil organic carbon (SOC), soil total nitrogen (STN)



and soil pH, can also affect Rs in that they can directly or indirectly affect substrate quality
and quantity, which strongly control soil microbial activity and heterotrophic respiration
(Ryan and Law, 2005; Chen et al., 2010a, 2014; Song et al., 2014). Additionally, biotic
factors including decomposer microbes and plant can directly influence soil respiration via
heterotrophic and autotrophic respiration, respectively (Ryan and Law, 2005; Bahn et al.,
2010). Previous studies have shown that Rs increased with total, aboveground and
belowground net/gross primary production, aboveground biomass (AGB) and
belowground biomass (BGB), leaf area index (Raich and Schlesinger, 1992; Hibbard et al.,
2005; Bahn et al., 2008; Chen at al., 2014; Zhao et al. 2017).

Similarly, the temperature sensitivity of Rs is also largely regulated by both biotic and
abiotic factors. Several studies have showed that climatic factors had strong controls on
the spatial variation of $Q_{10}$, and $Q_{10}$ generally decreased with mean annual temperature
(MAT) and mean annual precipitation (MAP) (Raich and Schlesinger, 1992; Kirschbaum,
2000; Peng et al., 2009; Song et al., 2014). In terms of geographical variables, latitude and
altitude can also indirectly influence $Q_{10}$ via controlling MAT and MAP (Song et al., 2014;
Xu et al., 2015). In addition to climatic and geographical variables, $Q_{10}$ could be affected
by other factors, such as plant biomass, soil pH, SOC and STN, which can directly
influence microbial activity, substrate availability and nutrient supply (Zhou et al., 2009;
Song et al. 2014; Zhao et al., 2017).

Grasslands in China cover 29–41% of its total land area (Shen et al., 2016) and have
significant effects on regional climate and carbon cycle (Ni, 2002). As known to all,
grasslands are widely distributed throughout China, and the different climate gradients and
landforms in China support a number of grassland types, including tropical, warm,
temperate, and alpine grassland, etc. (Chen et al., 2002; Shen et al., 2016). Specifically,
the temperate arid and semi-arid grasslands in Inner Mongolia, and the alpine meadow and
steppe in Qinghai-Tibet Plateau comprise the main body of temperate and alpine
grasslands, respectively (Shen et al., 2016). In the past two decades, a large number of
case studies on Rs have been widely conducted in grasslands across China. However, few
have been included in global Rs and $Q_{10}$ syntheses (Raich and Schlesinger, 1992; Wang et
al., 2010; Bond-Lamberty and Thomson, 2010; Chen et al., 2014; Hursh et al., 2017),
largely because most studies were published in Chinese journals. Given that diverse
grassland types, especially alpine grasslands distributed in China, Rs and $Q_{10}$ may vary



among grassland types due to the differences in abiotic and biotic factors, and the patterns
of Rs and $Q_{10}$ across Chinese grasslands may differ from global terrestrial ecosystems and
grasslands. However, how the spatial variation of Rs and $Q_{10}$ varies with abiotic and biotic
factors across Chinese grasslands and their differences among grassland types still remain
poorly understood.

In this study, we synthesized all the available data relating to Rs and $Q_{10}$ in grasslands
across China. Our main objectives were to: (1) analyze the spatial patterns of Rs and $Q_{10}$
across various grassland ecosystems in China; (2) compare the differences in Rs and $Q_{10}$
among grassland types; (3) identify how abiotic and biotic factors drive Rs and $Q_{10}$ among
sites on the regional scale, including geographic variables, climatic factors, soil properties
and biotic factors; and (4) compare the Rs and $Q_{10}$ in Chinese grasslands with those from
previous syntheses on the global and regional scale.

## 2 Materials and Methods

### 2.1. Data collection

Peer-reviewed journal articles and published theses (including available online) before 1
December, 2017 were searched using Web of Science and China National Knowledge
Infrastructure (CNKI, available online: http://epub.cnki.net) with the following search
term combinations: (soil respiration OR soil $CO_2$ flux OR soil $CO_2$ efflux OR soil $CO_2$
emission OR soil carbon flux OR soil carbon efflux OR soil carbon emission) AND
(grassland OR steppe OR meadow OR grass). Additional searches with the same
keywords were conducted on ScienceDirect (Elsevier Ltd., Amsterdam, Nederland),
Springer Link (Springer International Publishing AG, Berlin, Germany), and Wiley Online
Library (John Wiley & Sons Ltd., Hoboken, USA). Furthermore, previous global and
regional syntheses on the similar topic were also screened to check Chinese grassland
data, such as Peng et al. (2009), Wang and Fang (2009), Bond-Lamberty and Thomson
(2010), Wang et al. (2010) and Chen at al. (2010, 2014).

To ensure data consistency and accuracy, the following six criteria were applied to select
appropriate studies: (1) experimental studies were conducted in the field; (2) experiments
with the treatments of nitrogen (fertilizer) addition, increased or decreased precipitation,
elevated $CO_2$, simulated acid rain, clipping, and grazing were removed; (3) the study must
contain soil respiration or $Q_{10}$ with a clear record of grassland type and experimental
duration; (4) the investigation time for measuring Rs was not less than twelve months so
that the annual Rs can be obtained, and modeled annual Rs based on the relationships
between Rs rate and temperature were not included; (5) the investigation time for
estimating $Q_{10}$ value was not less than four months; and (6) $Q_{10}$ values were calculated by
the van't Hoff equation (Van's Hoff, 1898).
$$SR = \alpha \times \exp(\beta \times T) \tag{1}$$
where $SR$ is the measured soil respiration rate, $T$ is the measured soil temperature at the
given depth, and coefficient $\alpha$ and $\beta$ are fitted parameters. The $Q_{10}$ values were calculated
as:
$$Q_{10} = \alpha \times \exp(\beta \times T) \tag{2}$$

Several studies measured Rs and its temperature sensitivity at different years, and then
these Rs and $Q_{10}$ values were averaged across years. In this case, only the highest $R^2$ was
extracted if more than one coefficient of determination ($R^2$) values of $Q_{10}$ were available in
the same study. In addition, the annual $Q_{10}$ value was selected in our database if the
growing season, non-growing season, and annual $Q_{10}$ values were available. Within these
constraints, 54 measurements of annul Rs rate and 171 estimates of $Q_{10}$ value were
obtained from 108 published experimental studies across Chinese grasslands (Table S1).
Our database contained 14 variables associated with Rs, including annual Rs, growing and
non-growing season Rs and their proportions to annual Rs, the proportion of autotrophic
and heterotrophic respiration to annual Rs, and the $Q_{10}$ of Rs. The $Q_{10}$ values were divided
into five types based on the soil temperature at different depths (ST0, soil surface
temperature; ST5, soil temperature at 5 cm; ST10, soil temperature at 10 cm; ST15, soil
temperature at 15 cm; and ST20, soil temperature at 20 cm) for the same site. In one study,
the $Q_{10}$ was derived by soil temperature at the depth of 6 cm, and then it was treated as
$Q_{10\text{-ST5}}$ because of little difference in soil temperature between 5 cm and 6 cm.

In most of publications, the Rs, $Q_{10}$ and its $R^2$ of the model were presented, and they were
incorporated into our database directly. The Rs, $Q_{10}$ and $R^2$ values were recalculated
according to the available information if these values were not directly provided in some
publications. The growing season, non-growing season and annual Rs were obtained by
interpolating measured Rs rate between respective sampling dates for each seasonal



measurement period of the year, and then computing the sum to obtain the corresponding
values (Frank and Dugas, 2001; Sims and Bradford, 2001) as follows:
$\quad CSR = \Sigma\, (\Delta t_k \times F_{m,k})$ (3)
where $CSR$ is cumulative soil respiration during the season, $\Delta t_k\ (= t_k - t_{k-1})$ is the time
interval between each field measurement within the season, and $F_{m,k}$ is the average Rs rate
over the interval $t_k - t_{k-1}$.

In addition, for each study site, we also recorded other supporting information from the
original publications, including grassland type, geographic variables (longitude, latitude
and altitude), climatic factors (MAT and MAP), soil properties (soil pH, SOC and STN),
and biotic factors (microbial biomass carbon (MBC), AGB and BGB). Missing climatic
information were obtained using NASA Surface meteorology and Solar Energy-Location,
and the other missing information were obtained from the related references according to
the study site and described experiment design. Several studies provided the soil organic
matter content, which was converted to SOC by multiplying a constant of 0.58. Given that
BGB were measured in different soil depths, only BGB measured in 0−40 and 0−50 cm
soil depths were selected because roots were mainly distributed in 0−50 cm and there were
minor difference between 0−40 and 0−50 cm. The distributions of selected experimental
sites were showed in Fig. 1.

**2.2. Data analysis**
In this study, grasslands were divided into five groups, including temperate typical steppe,
temperate meadow steppe, temperate desert steppe, alpine grassland, and warm-tropical
grassland. If grassland type was not provided directly, it was determined according to the
dominant plant species reported in selected publications and the Classification of
Grassland Ecosystem in China (Chen et al., 2002). Detailed statistical parameters for the
five grasslands were presented in Table S2.
One-way analysis of variance (ANOVA) was used to examine whether annual Rs and $Q_{10}$
values differed among grassland types or soil temperature measurement depths. In case of
homogeneity of variances, the least significant differences (LSD) test was applied;
otherwise, the Dunnett T3 test was applied. Paired-samples $t$-test was performed to
compare the differences between growing season and non-growing season Rs, between
autotrophic respiration and heterotrophic respiration, and the $Q_{10}$ values among different



measurement depths. The univariate regression analysis was used to identify the
relationships between annual Rs, $Q_{10}$, and a given biotic or abiotic factor mentioned
above, except for MBC because of its limited sample size. The stepwise linear regression
analyses were also performed to identify the comprehensive effects of environmental
variables (including latitude, altitude, MAT, and MAP as they were in one-to-one
correspondence) on annual Rs, and $Q_{10}$ derived by ST5 and ST10. Correlations among
these factors were calculated with the Pearson correlation coefficients. All statistical
analyses were performed using the software IBM SPSS Statistics 20.0 (IBM Corporation,
New York, USA).

## 203   3 Results

### 204   3.1 Soil respiration and its controlling factors

### 205   3.1.1. Patterns of annual soil respiration across Chinese grasslands

The annual Rs ranged from 122.9 to 2407.1 g C m$^{-2}$ yr$^{-1}$, with the total mean ($\pm$ SE) of
582.0 $\pm$ 57.9 g C m$^{-2}$ yr$^{-1}$. There were significant differences in annual Rs between
grassland types ($p < 0.001$), with the highest annual Rs in the warm-tropical grassland and
the lowest annual Rs in the temperate desert steppe (Table 1). The proportions of growing
season or non-growing season Rs varied slightly among different grassland types ($P >$
0.05), but the proportion of Rs in growing season was significantly higher than that in
non-growing season ($p < 0.001$). Overall, growing season and non-growing season Rs
consisted of 78.7% and 21.3% of the annual Rs, respectively, across all grasslands in
China (Table 1). In addition, growing season Rs was significantly positively correlated
with the annual Rs based on linear regression model ($r^2 = 0.923$, $p < 0.001$, Fig. S1). At
the annual scale, the mean contribution of heterotrophic respiration to Rs was 72.8%
across Chinese grasslands, which was significantly larger than that of autotrophic
respiration with the mean of 27.2% ($p < 0.01$, Fig. S2).

### 220   3.1.2. Spatial controls of abiotic and biotic factors over soil respiration

In the univariate linear regressions, annual Rs significantly increased with MAT, MAP
SOC, and AGB across all grasslands in China, but decreased with latitude, altitude, and



soil pH ($p < 0.05$, Fig. 2). In contrast, annual Rs did not correlate well with STN and BGB
($p > 0.05$). The single factor of latitude, MAT, MAP, SOC, soil pH, and AGB accounted
for 25.7%, 22.4%, 31.3%, 30.2%, 20.6%, and 36.1% of the spatial variation of annual Rs,
respectively (Fig. 2). In addition, only the variable of MAP was selected in the analysis of
stepwise linear regression, indicating that MAP was the primary factor controlling the
spatial variation of annual Rs in Chinese grasslands.

**3.2 Temperature sensitivity of soil respiration and its controlling factors**
**3.2.1 Distributions of $Q_{10}$ values and its coefficient of determination**
Most of the $Q_{10}$ values (83.0%) were distributed between 1.5 and 3.8. However, the
distributions of $Q_{10}$ values derived by the five temperature types were different (Fig. 3a-e).
The largest relative frequency for $Q_{10\text{-ST5}}$ and $Q_{10\text{-ST10}}$ values was within the range of 1.5 to
3.0 (68.5%) and 1.5 to 3.5 (83.1%), respectively, while that of $Q_{10\text{-ST0}}$ was mainly within
1.0–2.0 (88.2%, Fig. 3). In addition, the distribution of $Q_{10\text{-ST15}}$ and $Q_{10\text{-ST20}}$ were relatively
uniform (Fig. 3d and e).

Similarly, the distributions of $R^2$ for $Q_{10}$ derived by the five temperature types also
differed from each other (Fig. 3f-g). The $R^2$ values for $Q_{10\text{-ST5}}$ and $Q_{10\text{-ST10}}$ were mainly
distributed in 0.6–0.9 and 0.5–0.7, respectively, while those for $Q_{10\text{-ST15}}$ and $Q_{10\text{-ST20}}$ were
mainly distributed in 0.3–0.6. The $R^2$ value for $Q_{10\text{-ST0}}$ was distributed uniformly (Fig. 3f).
Overall, only 35.6% $R^2$ values for $Q_{10}$ were within the range of 0.7–1.0.

**3.2.2 Patterns of $Q_{10}$ values across Chinese grasslands**
Across all grasslands, the overall $Q_{10}$ values ranged largely from 1.03 to 8.13, with the
mean ($\pm$ SE) of 2.60 $\pm$ 0.08. Specifically, the mean ($\pm$ SE) of $Q_{10}$ values derived by ST0,
ST5, ST10, ST15, and ST20 was 1.65 $\pm$ 0.08, 2.80 $\pm$ 0.14, 2.56 $\pm$ 0.12, 2.64 $\pm$ 0.33, and
2.81 $\pm$ 0.31, respectively (Fig.3 a-e). Paired $t$-test demonstrated that $Q_{10}$ significantly
differed between two adjacent depths in the top 15 cm soil ($P < 0.05$), whereas no
difference occurred below 15 cm depth ($p > 0.05$; Fig. 4). Generally, the overall $Q_{10}$ and
paired $Q_{10}$ increased with soil temperature measurement depth (Fig. 4; Fig. S3). In terms
of grassland types, there were significant differences for $Q_{10}$ derived by ST5 and ST10



among grassland types, respectively ($p < 0.05$, Fig. 4b and c). For $Q_{10}$ derived by ST5, it
was highest in alpine grassland, while for $Q_{10}$ derived by ST10, the highest value was in
warm-tropical grassland. In addition, $Q_{10}$ values derived by ST0, ST15 and ST20 were not
enough to meet the demand of statistical analysis, so their differences among grassland
types were not examined.

**3.2.3 Spatial controls of environmental factors over $Q_{10}$**
The relationships of $Q_{10\text{-ST5}}$ and $Q_{10\text{-ST10}}$ with abiotic and biotic factors were presented in
Fig. 5. Among these abiotic and biotic factors, $Q_{10\text{-ST5}}$ correlated well with latitude,
altitude, SOC, AGB and BGB ($P < 0.05$, Fig. 5). In contrast, $Q_{10\text{-ST10}}$ significantly
correlated with MAP and SOC ($P < 0.05$, Fig. 5). In addition, only three factors including
altitude, MAP and MAT were selected in the analysis of stepwise linear regression,
indicating that they interactively affected $Q_{10\text{-ST5}}$, and accounted for 26.0% of the spatial
variation of $Q_{10\text{-ST5}}$ across Chinese grasslands (Table S3).

**4 Discussion**
**4.1 Spatial patterns and controlling factors of annual soil respiration**
**4.1.1 Annual soil respiration among grassland types**
Significant differences among the five grasslands suggested grassland type had significant
influence on annual Rs ($p < 0.001$, Table 1), which might be mainly attributed to the
differences in AGB, BGB and microbial activity across various grassland types. Previous
incubation experiments showed microbial respiration positively correlated with microbial
biomass (Colman and Schimel, 2013; Ding et al., 2016), indicating grasslands with higher
MBC would have larger heterotrophic respiration. Meanwhile, regional study suggested
that microbial biomass was closely increased with MAP in grasslands (Chen et al., 2016b),
which was also found in this study. Altogether, these suggested that the regions with high
MAP would have larger heterotrophic respiration. Additionally, previous study
demonstrated that both AGB and BGB increased with MAP across Chinese grasslands
(not including warm-tropical grasslands) (Ma et al., 2014). Therefore, autotrophic
respiration would be higher in the grasslands with high biomass. Collectively, the



grasslands with high MAP would have relatively higher Rs rate. Our results also showed
this trend that mean annual Rs in each of the four grassland types increased significantly
with MAP ($p < 0.01$, Fig S4).

**4.1.2 Controls of environmental factors on annual Rs**
Across Chinese grasslands, annual Rs were strongly related to latitude, MAT and MAP,
which were consistent with previous results obtained from global terrestrial ecosystems
(Raich and Schlesinger, 1992; Raich and Potter 1995; Chen at al., 2014), global grasslands
(Wang and Fang, 2009), and Chinese forests (Song et al., 2014; Xu et al., 2015). As a key
factor controlling climate conditions on the regional and global scale, latitude could
significantly influence Rs by affecting climatic variables (Song et al., 2014). Our study
showed that MAT and MAP decreased closely with latitude ($p < 0.001$, Table S3),
indicating that latitude is an indirect factor affecting annual Rs on the large scale.

In addition, spatial variations of annual Rs were also controlled by soil properties, such as
SOC and soil pH. The relationships between annual Rs and SOC as well as pH were also
observed in global, regional and local terrestrial ecosystems (Chen et al., 2010b, 2014;
Song et al., 2014; Xu et al., 2016). Since Rs involves the process of converting organic
carbon into inorganic carbon, the soil $CO_2$ emission is untimely determined by the supply
of C substrate (Wan et al., 2007). Additionally, soil pH can directly regulate the activities
of microbes and C-acquiring enzymes (Turner, 2010). In neutral and alkaline soils,
microbial biomass tended to decrease with soil pH (Ding et al., 2016). Therefore, this led
to a negative correlation between Rs and soil pH in Chinese grasslands because most of
grasslands in China are distributed in neutral and alkaline soils. Further, Chen et al.
(2010b) demonstrated that annual Rs significantly increased with soil total nitrogen on the
global scale. Meanwhile, some case studies revealed the similar relationship between
growing season Rs and soil total nitrogen among different grassland types and vegetation
communities (Chen et al., 2010a; Wang et al., 2015; Xu et al., 2016) on the local scale,
while annual Rs did not correlate well with STN in this study. Altogether, these results
suggested that the effect of soil total nitrogen on Rs depended on plant growth period,
vegetation type, and spatial scale. Therefore, how STN influence Rs across Chinese
grasslands on the regional scale should be further studied.



Furthermore, as the source of autotrophic respiration, BGB can directly influence Rs,
which has been observed in ecosystems on global and local scale (Chen at al., 2010a,
2014). However, no significant correlation between BGB and Rs was observed in the
present study, which might be attributed to the limited sample size ($n = 20$) and the
uncertainty in measuring BGB (due to inconsistent or insufficient sampling depth). In
grassland ecosystems, BGB generally increased with AGB (Ma et al., 2014), and this
relationship was also observed in this study ($p < 0.10$, Fig. S5). Therefore, given the
significant correlation between AGB and Rs in Chinese grasslands (Fig. 2), BGB may also
have the potential to control annual Rs across Chinese grasslands, although this should be
further investigated based on accurate quantification of BGB and Rs across a large number
of sites.

**4.2 Spatial patterns and controlling factors of $Q_{10}$ values**
**4.2.1 $R^2$ for $Q_{10}$ in Chinese grasslands**
In this study, only 37.3% of $R^2$ for $Q_{10}$ was larger than 0.7, indicating that most of the
seasonal variation of Rs rate cannot be well explained by soil temperature using the van't
Hoff equation (Eq. 2). Compared with the results obtained from Chinese forests (Xu et al.,
2015), the van't Hoff equation (Eq. 2) was not very suitable to describe the relationships
between Rs rate and soil temperature in most of Chinese grasslands. This might be
associated with the difference in soil moisture between these two ecosystems because
besides temperature, soil moisture may strongly influence the apparent $Q_{10}$ (Subke and
Bahn, 2010). Previous studies have suggested that in humid and semi-humid regions the
effect of soil moisture on Rs is weak, whereas in arid and semi-arid regions, Rs is
significantly influenced by soil moisture (Jia et al., 2006; Li et al., 2011; Wang et al.,
2014a, 2014b). Moreover, some studies showed that soil moisture and temperature had an
interactive effect on the seasonal variations of Rs rate (Davidson et al., 1998; Jia et al.,
2006; Wang et al., 2014b; Liu et al., 2016), indicating that the two-variable equations
could better explain the variation in Rs than single variable of temperature. Our results
also showed that, in general, $R^2$ for $Q_{10}$ closely increased with MAP ($P < 0.05$, Fig. S6),
indicating that the $R^2$ for $Q_{10}$ tended to be larger in the regions with abundant precipitation.
Collectively, for ecosystems (e.g., grassland and desert) in arid and semi-arid regions, Rs
could be better estimated by the combined factors of soil temperature and moisture. By



comparison, 46.6% of $R^2$ for $Q_{10\text{-}ST5}$ was distributed in 0.7–1.0, which was higher than
those derived by soil temperature at other depths, suggesting that the seasonal variation of
Rs can be better explained by soil temperature at the depth of 5 cm across Chinese
grasslands.

**4.2.2 $Q_{10}$ among soil depths and grassland types**
In Chinese grasslands, the estimated $Q_{10}$ generally increased with soil temperature
measurement depth, which was consistent with previous synthesis study about Chinese
ecosystems (Peng et al., 2009). The differences for $Q_{10}$ among measurement depths might
be due to the seasonal amplitudes of temperature at different soil depths (Pavelka et al.,
2007; Graf et al., 2008).

In terms of grassland types, the highest $Q_{10\text{-}ST5}$ was in the alpine grassland and the lowest
in the temperate desert steppe and typical steppe (Fig. 4). This difference could be
associated with soil properties and climatic conditions. For example, it is well known that
the alpine grasslands are usually distributed in high altitude regions (above 3000 m),
where the climate is relatively colder and SOC is relatively higher (Table S2). However,
the temperate desert steppes and typical steppes are mainly distributed in north China,
with relatively high MAT and low MAP that may lead to low $Q_{10}$. Moreover, as shown in
Fig. 4, the highest $Q_{10\text{-}ST10}$ occurred in warm-tropical grassland, which might be associated
with the abundant substrate supply in this grassland type because high substrate
availability can enhance apparent $Q_{10}$ of soil respiration (Davidson et al., 2006; Zhu and
Cheng, 2011).

**4.2.3 Controls of environmental factors on $Q_{10}$**
Generally, the $Q_{10}$ derived by either ST5 or ST10 did not correlate well with climatic
factors, which was inconsistent with previous results on the global and regional scale
(Chen and Tian, 2005; Peng at al., 2009; Wang et al., 2010; Song et al., 2014; Xu et al.,
2015). This suggested that the single factor of temperature or precipitation could not
critically control the variations of $Q_{10}$ in Chinese grasslands, which are mainly distributed
in arid and semiarid regions. In addition, the negative correlation between latitude and
$Q_{10\text{-}ST5}$ in Chinese grasslands was not in line with Chinese forests, in which positive



correlation was observed (Song et al., 2014; Xu et al., 2015). The difference might be that
alpine grasslands in China were mainly distributed in regions with low latitude but high
altitude. Previous studies and the present result indicated that $Q_{10}$ tended to be higher at
high altitude regions (Song et al., 2014; Xu et al., 2015).

Additionally, the positive relationships of $Q_{10\text{-ST5}}$ with SOC, AGB and BGB indicated that
soil properties and plant biomass can also profoundly influence the spatial variation of
$Q_{10}$. Previous studies suggested higher plant biomass and SOC can lead to more substrate
supply for soil respiration and then result in higher $Q_{10}$ values, because apparent $Q_{10}$
increased with increasing substrate availability (Gershenson et al., 2009; Zhao et al.,

391  2017).


The extremely low $R^2$ value for the relationship of $Q_{10}$ with abiotic factors suggested that
the spatial variation of $Q_{10}$ in Chinese grasslands cannot be well explained by a single
factor. Therefore, the variation of $Q_{10}$ in Chinese grasslands should be controlled by
multiple factors due to the complex and diverse environments among grasslands on the
large scale. Stepwise linear regression analysis also demonstrated that latitude, MAP and
MAT had the comprehensive effects on the spatial variation of $Q_{10\text{-ST5}}$. Additionally, both
univariate and multiple regression analyses demonstrated that generally there were no
significant relationships between $Q_{10\text{-ST10}}$ and abiotic and biotic factors, indicating that the
$Q_{10\text{-ST10}}$ did not have clear spatial pattern. Therefore, the variation of $Q_{10\text{-ST10}}$ might be
controlled by other factors, and should be further studied.

**4.3 Comparisons of Rs and $Q_{10}$ between Chinese grasslands and the global ecosystems**
**4.3.1 Comparisons of annual Rs**
The annual Rs varied largely within and among the grassland types across China (Table 1),
with the mean value of 582.0 g C m$^{-2}$ yr$^{-1}$, which was much lower than that in global
terrestrial ecosystems (Table 2). Similarly, the mean annual Rs rate in Chinese grasslands
was also much lower than that in Chinese forests. For these global results, the main
biomes in their dataset were forests, which had relatively higher Rs than grasslands.
Therefore, this would lead to the differences between Chinese grasslands, and Chinese
forests and global terrestrial ecosystems (Table 2). Compared with global grasslands, our



result was much lower or higher than the results obtained from Chen et al. (2010b, 2014)
and Wang and Fang (2009), but approximately consistent with Hursh et al. (2017). In
general, the mean annual Rs rate across Chinese grasslands was between the lowest and
highest Rs across global grasslands.

Across Chinese grassland types, the proportions of Rs in growing season ranged from
76.2–86.8%, which were 2.2–5.6 times higher than those in non-growing season. As a
whole, heterotrophic respiration contributed 72.8% of the annual Rs, 2.7 times of
autotrophic respiration, which was close to that of global terrestrial ecosystems and
grasslands (Wang and Fang, 2009; Chen et al., 2014) and Chinese forests (Song et al.,
2014). Generally, our findings and previous studies suggested that Rs during growing
season and heterotrophic respiration was an important part of the annual Rs, respectively,
and should be given enough attention.

**4.3.2 Comparisons of $Q_{10}$**
The overall mean $Q_{10}$ of 2.60 derived by soil temperature at all measurement depths was
similar to global terrestrial ecosystems with 2.40 and 2.54 (Raich and Schlesinger, 1992;
Lenton and Huntingford, 2003). The $Q_{10}$ derived by ST5 varied from 1.39 to 8.13, with the
mean of 2.80, which was higher than that of global and Chinese terrestrial ecosystems,
Chinese forests, especially higher than that of global grasslands (Table 2). The difference
may be partly due to the distribution of grasslands in China and the grassland types.
Chinese grasslands are mainly distributed in the high latitude (temperate grassland) and
high altitude (Qinghai-Tibet Plateau alpine grassland) regions, and $Q_{10}$ takes relatively
higher values in cold regions than in warm regions (Chen and Tian, 2005; Wang et al.,
2010). In addition, in this study, averaged $Q_{10-ST5}$ was highest in alpine grassland with the
mean of 3.30. However, there were no alpine grasslands in the global database.
Collectively, this may lead to higher $Q_{10}$ value in Chinese grasslands. In terms of $Q_{10}$
derived by ST10, the mean value for Chinese grasslands was close to Chinese terrestrial
ecosystems, but much lower than the global ecosystems (Table 2).

**4.4 Uncertainties**





In order to ensure data consistency and minimize the error, only field experiments in
accordance with the six aforementioned criteria were selected. However, the inter-annual
variation in Rs and $Q_{10}$ might be very large for grassland at one site, which was associated
with the variations in annual precipitation and mean temperature between two adjacent
years (Peng et al., 2014; Wang et al., 2016). Therefore, the inter-annual variation of Rs
would impact the accuracy of the results. Additionally, three methods including static
closed chamber, dynamic closed chamber, and alkali absorption were widely applied to
measure Rs in the selected experiments, and previous studies have suggested that
measurement methods affected the results of Rs rate and $Q_{10}$ value (Bekku et al., 1997;
Yim et al., 2002; Peng et al., 2009). However, in this study, there were no significant
differences for $Q_{10\text{-ST5}}$ and $Q_{10\text{-ST10}}$ among the three measurement methods (Fig. S7).
Given that only one sample of annual Rs was measured by alkali absorption, therefore the
effects of measurement methods on Rs could be neglected. Including the data measured by
the AA method in our synthesis does not meaningfully change the results of Rs and $Q_{10}$.

In this study, the selected experiments were mainly conducted in temperate and alpine
grasslands, so the limited data obtained from desert, tropical and subtropical grasslands
might lead to some uncertainties in these ecosystems. Moreover, grassland management
practices such as land use/cover change, intensity and pattern of livestock grazing, and
fencing can have significant effect on soil carbon emission (Chen at al., 2013; Zhang et
al., 2015b; Chen et al., 2015; Chen at al., 2016a). In the past three decades, several
ecological projects relating to grassland have been implemented in China, and have
observably increased the grassland area and altered the land cover (Zhang et al., 2015a).
To some extent, these changes can also impact our findings.
**5 Conclusion**
Chinese grasslands cover vast area, have high spatial heterogeneity, and include various
grassland types. By synthesizing all the available data relating to Rs and $Q_{10}$, we analyzed
their spatial patterns and driving factors in grasslands across China. Our results showed
that Rs and its temperature sensitivity varied largely within and among grassland types,
with the mean annual Rs and $Q_{10}$ of 582.0 g C m$^{-2}$ yr$^{-1}$ and 2.60, respectively. MAT, MAP,
and SOC all significantly positively affected annual Rs, whereas both latitude and soil pH
negatively affected annual Rs. The Rs during growing season and heterotrophic respiration



were the major component of annual Rs, contributing 78.7% and 72.8% of the annual Rs,
respectively. The altitude, MAP and MAT were the dominant factors and accounted for
26.0% of the variation of $Q_{10\text{-ST5}}$ across Chinese grasslands. These findings should
advance our understanding of the spatial variation and environmental control of soil
respiration and $Q_{10}$ across Chinese grasslands, and also improve our ability to predict soil
carbon efflux under climate change on the regional scale.

**Acknowledgement**

This research was financially supported by the National Key Research and Development
Program of China (2017YFC0503903 and 2016YFC0502006), the National Natural
Science Foundation of China (31621091 and 31622013), and the Key Science and
Technology Plan Projects of Tibet Autonomous Region (Z2016C01G01/08-004 and
Z2016C01G01/03). We thank all the scientists whose data and work were included in this
synthesis.

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



**Table 1** The annual soil respiration (Rs) and the proportions of growing season,
non-growing season Rs to annual Rs in different grassland ecosystems across China.

| Grassland types | N | Annual Rs (g C m$^{-2}$ yr$^{-1}$) | | | Rs proportion (%) | |
|---|---|---|---|---|---|---|
| | | Mean ± SE | Min. | Max. | Growing season | Non-growing season |
| Temperate typical steppe | 16 | 371.3±94.8 a | 122.9 | 1670.0 | 79.6±2.9 a | 20.4±2.9 a |
| Temperate meadow steppe | 6 | 442.1±83.4 a | 218.8 | 784.7 | 86.8±2.7 a | 13.2±2.7 a |
| Alpine grassland | 20 | 581.5±62.3 a | 246.3 | 1161.1 | 77.3±2.5 a | 22.7±2.5 a |
| Warm-tropical grassland | 12 | 933.6±161.8 b | 428.8 | 2407.1 | 76.2±2.5 a | 23.8±2.5 a |
| Total | 54 | 582.0±57.9 | 122.9 | 2407.1 | 78.7±1.5 A | 21.3±1.5 B |

There was no sample for temperate desert steppe, so the data was not presented in this
table. The different lowercase letters in each column indicate the significant difference at
$P = 0.05$, and different uppercase letters indicate the significant difference between
growing and non-growing season at $P = 0.001$. N: number of samples.



**Table 2** The comparisons of annual soil respiration and $Q_{10}$ between Chinese grasslands
and other syntheses.

| Scope | Annual Rs (g C m$^{-2}$ yr$^{-1}$) | $Q_{10\text{-ST5}}$ | $Q_{10\text{-ST10}}$ | Reference source |
|---|---|---|---|---|
| Global terrestrial ecosystems | 910.0 (657) | | | Chen at al., 2010 |
| | 870.0 (1195) | | | Chen at al., 2014 |
| | 791.2 (1741) | | | Hursh et al., 2017 |
| | | 2.40 (77) | 3.10 (46) | Wang et al., 2010 |
| Global grasslands | 448.9 (46) | 2.13 (41) | | Wang and Fang, 2009 |
| | 745.0 (179) | | | Chen at al., 2010 |
| | 840.0 (113) | | | Chen at al., 2014 |
| | 599.1 (163) | | | Hursh et al., 2017 |
| Chinese terrestrial ecosystems | | 2.03 (64) | 2.61 (33) | Peng et al., 2009 |
| Chinese forests | 919.7 (139) | 2.46 (107) | | Song et al., 2014 |
| | | 2.51 (145) | | Xu et al., 2015 |
| Chinese grasslands | 582.0 (54) | 2.80 (73) | 2.56 (59) | This study |

The numbers in parentheses represent the number of samples.






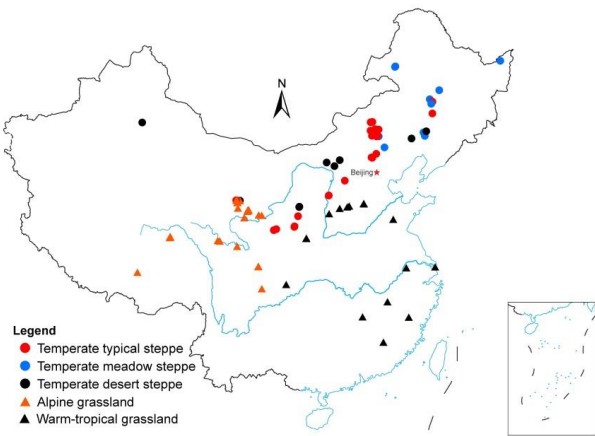

**Figure 1.** The site location of soil respiration studies selected in this study across
Chinese grasslands.





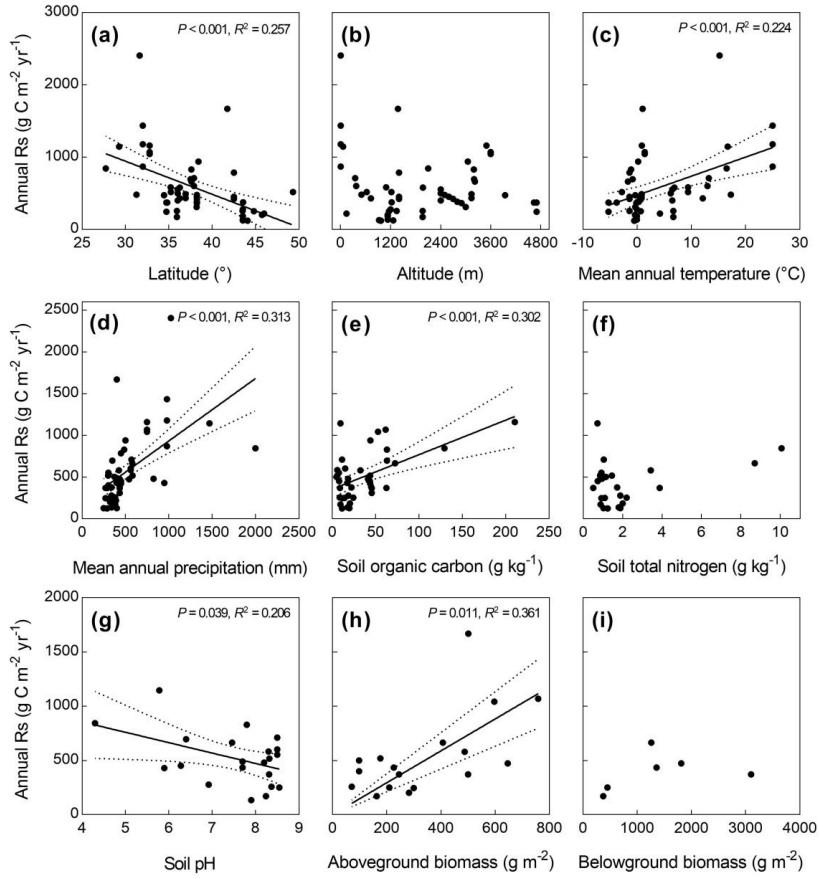


**Figure 2.** Relationships of annual soil respiration (Rs) with abiotic and biotic factors.

The dash lines represent the 95% confidence interval.






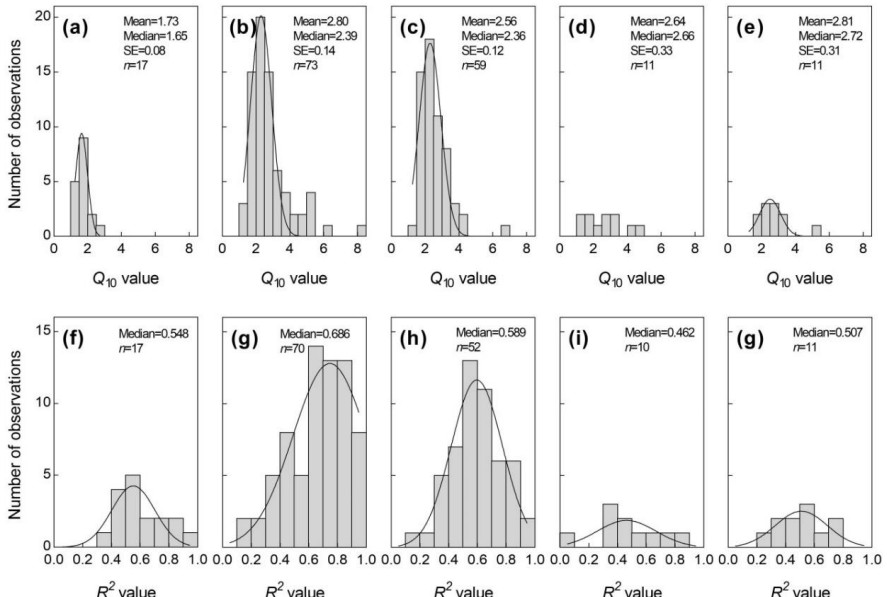


**Figure 3.** Histogram plots for $Q_{10}$ values (a-e) and its coefficient of determination ($R^2$)

for $Q_{10}$ (f-j) across Chinese grasslands. (a) and (f): soil surface temperature; (b) and (g):

soil temperature at the depth of 5 cm; (c) and (h): soil temperature at the depth of 10 cm;

(d) and (i): soil temperature at the depth of 15 cm; (e) and (j): soil temperature at the

depth of 20 cm. $n$ represents the number of samples.

677





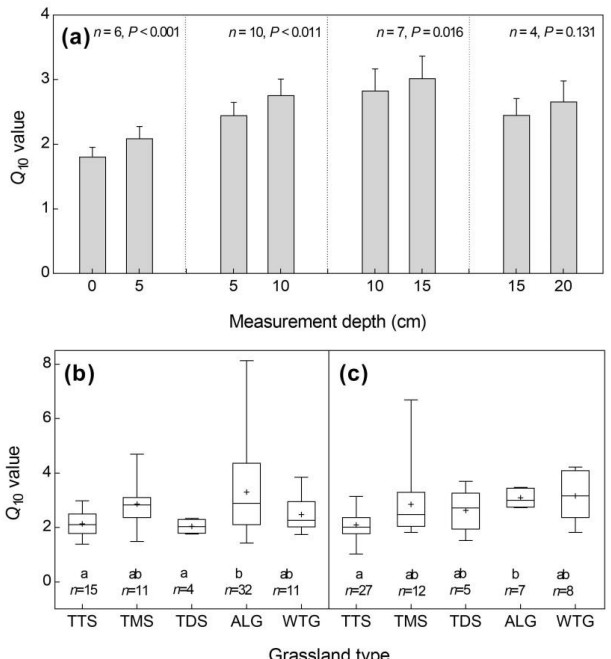

678

**Figure 4.** Comparisons $Q_{10}$ values among soil temperature measurement depths (a) and among grassland types (b, c). (a) $Q_{10}$ values derived by soil temperature at the depth of 0, 5, 10, 15, and 20 cm, respectively. (b) $Q_{10}$ values derived by soil temperature at the depth of 5 cm. (c) $Q_{10}$ values derived by soil temperature at the depth of 10 cm. TTS, TMS, TDS, ALG, and WTG represent temperate typical steppe, temperate meadow steppe, temperate desert steppe, alpine grassland, and warm-tropical grassland, respectively. In the box plot, the "+" represent mean values, horizontal lines inside box represent medians, box ends represent the 25th and the 75th quartiles, vertical lines represent 2.5th and 97.5th percentiles, hollow circles represent outliers, and $n$ represents the number of samples. Error bars represent standard errors. Different lowercase letters indicate significant differences among soil depths or grassland types at $P = 0.05$.

690





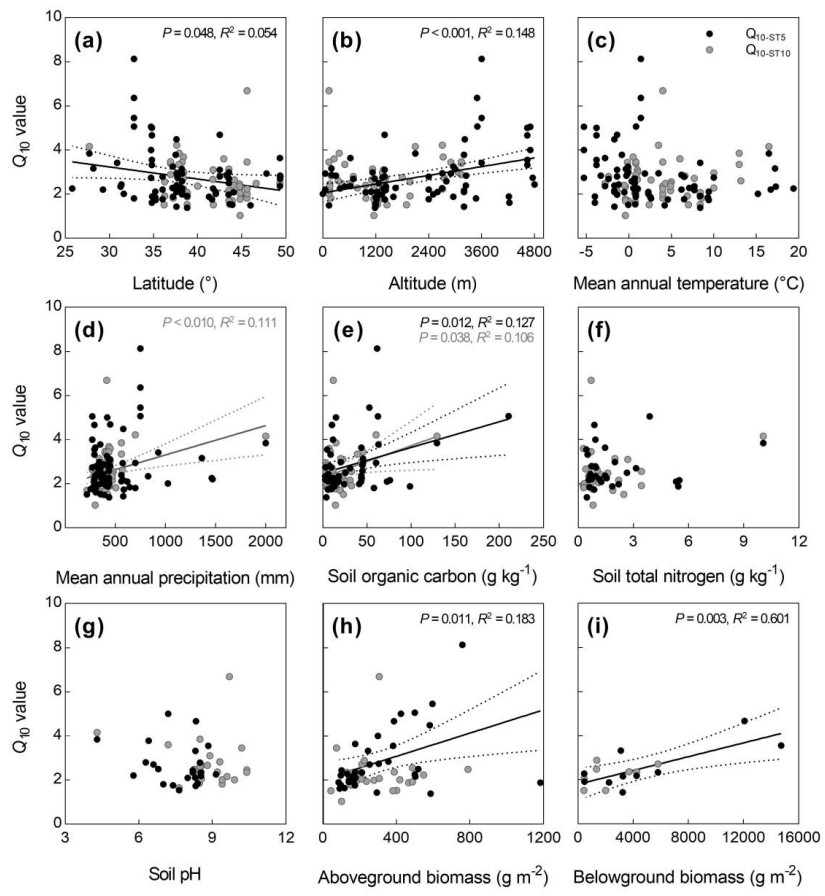

691

**Figure 5.** Relationships of the $Q_{10}$ derived by soil temperature at the depth of 5 cm with

abiotic and biotic factors.

694