# Peer review of "Patterns of soil respiration and its temperature sensitivity in"

_Biogeosciences, 2018_

## Referee Comment (RC1) · Anonymous Referee #1 · 2 Apr 2018

This manuscript made a contribution by compiling published data of soil respiration and temperature sensitivity related to soil respiration from five types of Chinese grasslands. The spatial extend of the dataset covers a large region. The temporal extend of the dataset is at the annual scale. It seems that the majority of the data points in this dataset have not been integrated into any published synthesis yet. Some aspects of the manuscript deserve attention. The authors carried out some basic correlation analyses on this dataset, and found some inconsistencies as compared with results in some published reports. One inconsistency was the correlation between annual soil respiration rate (Rs) and total soil nitrogen content (or total soil carbon content, because soil C and N tend to go together). As normally expected, most published reports showed

highly significant correlation between Rs and soil C &N, but not this manuscript. The actual causes of this inconsistency were unclear. Another inconsistency was that the manuscript did not find any significant correlations between climatic variable (e.g., temperature and precipitation) and Q10 values measured at 5cm or 10cm depth, which is in contrast to published results. Again, clear causes of this inconsistency were not offered. As the authors stated in the manuscript, the soil respiration in this context has two main components: autotrophic respiration of plant roots, and heterotrophic respiration of soil microbes. Therefore, the soil respiration should be controlled by both plant-related variables and soil-related variables. But unfortunately, there were only 7 data points that have autotrophic and heterotrophic respiration measured separately (and probably using questionable methods). Consequently, Rs and Q10 data could not be discussed in relations to plant-related variables and soil-related variables. Furthermore, these Q10 values were calculated using the seasonally changing temperature data which often highly co-vary with plant growth (therefore, the seasonal increase of root respiration). As a result, the seasonal increase of root respiration would contribute to abnormally high Q10 values. This key aspect definitely needs authors' attention. Changes in the Introduction, Materials and Methods, and Discussion sections are required accordingly.

The followings are minor editorial comments: Line 25: 'latitude and' should be removed here. These geographic features (e.g., latitude, longitude, altitude or elevation) may be used as proxies for temperature or precipitation in data analysis only when temperature or precipitation data were not available. So authors should consider eliminate all parts of the manuscript that use these geographic features in statistical analyses and any related discussion. Line 28: The % heterotrophic respiration was only based on 7 data points, therefore, should not be in the abstract. Similarly, if the authors really want to make the "key" point of growing season vs. non-growing season, they should have given clear descriptions about how the separation was done accurately and reliably. Lines 29-31: This sentence needs a re-write so that the meaning becomes clear. Line 33: Remove the sentence about latitude and longitude here (the reason is given at line

25). Lines 35-38: Authors need to substantiate about 'how have they advanced the understanding' here. Line 53: "on the large scale'? Do you really want to 'step' on the large scale by the wall? My guess is that you really want to state: 'at a large scale' here. This correction should be made throughout the entire manuscript. Lines 67-68: Move the "and" to the place before the last part of the sentence, before "leaf area index" Line 83: "As known to all, . . ." The sentence is awkward. Line 133 and line 137: How could equations (1) and (2) have the same right sides? Also, what is the time factor for the T here? Is it measured at hourly, daily, weekly or annually time period? Line 155: Please define the "R-square and the model" here. Line 174: Why using "a constant of 0.58" here? I think it should be 0.5 now (see Pribyl 2010, Geoderma 156: 75–83). Line 263: "Q10-ST10" is not shown by Figure 5. Did you mean Q10-ST5? Line 267: Not "Table S3", should be Table S4. Line 302: "untimely" should be 'ultimately' Lines 308-315: The discussion here is unclear. Line 320: "n=20" here, but there were only 6 dots in the figure? Lines 331-352: These low R-square values could be a serious problem for this manuscript. How did you deal with this issue? Lines 405-425: This section is really rough. The quality of the discussion needs improvement. Lines 453-457: To me, Fig.7 actually showed huge differences between those three methods. Lines 471-473: The sentence structure is problematic. Lines 468-481: The Conclusion really needs lots of improvement.

---

## Referee Comment (RC2) · Anonymous Referee #2 · 11 Jun 2018

In this paper, the authors used published data to analyze the variations of soil $CO_2$ respiration rates and their temperature sensitivity (Q10) across Chinese grasslands. Furthermore, their relationships with some abiotic and biotic factors were analyzed. The results could advance the understanding of the variation and control factors of soil $CO_2$ respiration rates and their temperature sensitivity (Q10).

The specific comments are as follows:

Line 72: shown

Line 137: Correct the equation 2

Line 148-151: The Q10 values were divided into five soil depth with different soil temperature

Line 178: shown

Line S2: add the measuring methods

Line 192, Fig. 4: Why choose paired sample t-test to analyze the significant differences of the Q10 among the different soil depths?

Line 209: there is no results for the temperate desert steppe in Table 1

Line 233 and Line 239: five soil depths

Line 248: 1.73±0.08

Line 267 Table S4

Line 271-286, most of the contents are descriptive and repeated with results

Line 364 relatively colder and higher than what?

Fig. 2,5:   indicate the n values for each regression analysis

Fig. 3 Line 675 (e)and (g)

Table S1: whar $R^2$ represent for? What the ranges of soil temperature and soil moisture?

Table S2: show the n values. Are there values of soil temperature and soil moisture?

Table S3: show the n values. Are there values of soil temperature and soil moisture?

Fig. S1, S5, S6: show the n values

Fig. S7: is data for method comparison from the same or similar sites? Otherwise, there may be many factors affect the annual Rs and Q10.

---

## Author Comment (AC1) · 15 Jul 2018

General comments This manuscript made a contribution by compiling published data of soil respiration and temperature sensitivity related to soil respiration from five types of Chinese grasslands. The spatial extend of the dataset covers a large region. The temporal extend of the dataset is at the annual scale. It seems that the majority of the data points in this dataset have not been integrated into any published synthesis yet. Some aspects of the manuscript deserve attention. The authors carried out some basic correlation analyses on this dataset, and found some inconsistencies as compared with results in some published reports. One inconsistency was the correlation between

annual soil respiration rate (Rs) and total soil nitrogen content (or total soil carbon content, because soil C and N tend to go together). As normally expected, most published reports showed highly significant correlation between Rs and soil C & N, but not this manuscript. The actual causes of this inconsistency were unclear. Another inconsistency was that the manuscript did not find any significant correlations between climatic variable (e.g., temperature and precipitation) and Q10 values measured at 5 cm or 10 cm depth, which is in contrast to published results. Again, clear causes of this inconsistency were not offered. Response: Thanks for the constructive comments. We show our response to the three main comments on the inconsistency between our results and previous studies.

The first inconsistency was the correlation between annual soil respiration rate (Rs) and total soil nitrogen. In this study, we found that annual soil respiration did not significantly correlate with soil total nitrogen ($p = 0.10$, Fig. 2f), which was not consistent with previous results at the regional and global scale. Not surprisingly, we found that soil organic carbon was closely associated with soil total nitrogen ($p < 0.01$, Table S3). But, annual soil respiration increased closely with soil organic carbon ($p < 0.001$, Fig. 2e). The non-significant correlation between soil total nitrogen and annual soil respiration might be due to the limited sample size in soil total nitrogen compared to soil organic carbon (24 vs. 40), and/or due to the fact that soil total nitrogen might not well represent nitrogen availability for plants and microbes.

The second inconsistency was that this study did not find any significant correlations between climatic variables (i.e. mean annul temperature (MAT) and mean annual precipitation (MAP)) and Q10 values measured at 5 or 10 cm depth. This was not consistent with previously published results. But, we found Q10 measured at 5 or 10 cm soil depth was significantly decreased with increasing soil temperature, partly supporting the previous statement that Q10 tends to be higher in colder regions. Additionally, although the single factor of precipitation or temperature only explained a small proportion of the spatial variation of Q10, the combined factors of MAT and MAP, or soil

temperature and soil moisture, explained a significant proportion of the spatial varia-tion of Q10 across Chinese grasslands at regional scale (Table S4). Please see the discussion in section 4.2.3 Controls of environmental factors on Q10.

As the authors stated in the manuscript, the soil respiration in this context has two main components: autotrophic respiration of plant roots, and heterotrophic respiration of soil microbes. Therefore, the soil respiration should be controlled by both plant-related vari-ables and soil-related variables. But unfortunately, there were only 7 data points that have autotrophic and heterotrophic respiration measured separately (and probably us-ing questionable methods). Consequently, Rs and Q10 data could not be discussed in relations to plant-related variables and soil-related variables. Furthermore, these Q10 values were calculated using the seasonally changing temperature data which often highly co-vary with plant growth (therefore, the seasonal increase of root respiration). As a result, the seasonal increase of root respiration would contribute to abnormally high Q10 values. This key aspect definitely needs authors' attention. Changes in the Introduction, Materials and Methods, and Discussion sections are required accord-ingly. Response: Thanks for the constructive comments. When discussing annual soil respiration among grassland types, we analyzed autotrophic (root) respiration and het-erotrophic (microbial) respiration, respectively, for example, section 4.1.1 Annual soil respiration among grassland types. In addition, as the substrate of microbial decom-position, soil organic carbon (SOC) affects soil respiration. In addition, soil pH mainly controls heterotrophic respiration via regulating soil microbial activities. Therefore, the discussions related to SOC and pH were associated with plant-related variables and soil-related variables. But, the few samples (n = 7) from heterotrophic respiration and autotrophic respiration measured separately limited the in-depth discussions.

As you stated, the seasonal dynamics of plant growth affect root respiration and thereby seasonal Q10. At large scale, the seasonal amplitude of plant activity among different sites varied largely, which could affect the calculated seasonal Q10. Indeed, a previ-ous global synthesis study found that seasonal amplitude of plant activity fundamentally

dominates seasonal Q10 among different study sites compared with other environmental factors (Wang et al. 2010). But, in this study, we could not analyze the effects of seasonal variation of root respiration on Q10 due to the limited samples (n = 7) from autotrophic (root) respiration. In addition, the seasonal dynamics of plant growth at a given site might also affect the calculated Q10. In this study, our dataset included Q10 estimated at different time scale for measuring soil respiration. We categorized them into three types according to plant growth stage, including growing season Q10, non-growing season Q10, and annual Q10. In this case, we also conducted a one-way ANOVA analysis to examine the effects of measurement period (including growing season, non-growing season and annual scale) on Q10 derived by soil temperature at the depth of 5 and 10 cm, and found that measurement period did not significantly affect Q10 derived by soil temperature at the depth of 10 cm, but significantly affected Q10 derived by soil temperature at the depth of 5 cm. (Fig. S7). We have discussed this result in section 4.4 Uncertainties. Following your suggestions, we have revised the related content in the sections of Introduction, Materials and Methods (Section 2.2 Data analysis), and Discussion (section 4.2.3 Controls of environmental factors on Q10 and Section 4.4 Uncertainties).

Specific/Minor comments Line 25: 'latitude and' should be removed here. These geographic features (e.g., latitude, longitude, altitude or elevation) may be used as proxies for temperature or precipitation in data analysis only when temperature or precipitation data were not available. So authors should consider eliminate all parts of the manuscript that use these geographic features in statistical analyses and any related discussion. Response: Thanks for your good suggestion! We have eliminated all parts of the manuscript that use these geographic features in statistical analyses and any related discussion. In the revised manuscript, we added statistical analyses and discussion of soil temperature and soil moisture.

Line 28: The % heterotrophic respiration was only based on 7 data points, therefore, should not be in the abstract. Similarly, if the authors really want to make the "key" point

of growing season vs. non-growing season, they should have given clear descriptions about how the separation was done accurately and reliably. Response: Thanks for your good suggestions! We have eliminated the contents related to % heterotrophic respiration and % growing season in the abstract. In addition, we described how the growing season and non-growing season were defined. The growing season was from May to October, and the non-growing season was from November to April in the second year.

Lines 29-31: This sentence needs a re-write so that the meaning becomes clear. Response: Thanks for your good suggestion. We have re-written the sentence in the revised manuscript.

Line 33: Remove the sentence about latitude and longitude here (the reason is given at line 25). Response: Thanks for your good suggestions. We have removed all the sentences related to latitude and longitude in the revised manuscript.

Lines 35-38: Authors need to substantiate about 'how have they advanced the understanding' here. Response: Thanks for your good suggestion. We have revised the abstract and substantiated which understandings were advanced (line 41-45).

Line 53: "on the large scale"? Do you really want to 'step' on the large scale by the wall? My guess is that you really want to state: 'at a large scale' here. This correction should be made throughout the entire manuscript. Response: Thanks for your good suggestions. We have changed "on the large scale" to "at a large scale" throughout the entire manuscript.

Lines 67-68: Move the "and" to the place before the last part of the sentence, before "leaf area index". Response: Thank you. We have moved the "and" before "leaf area index".

Line 83: "As known to all …" The sentence is awkward. Response: Thank you. We have re-written the sentence.

Line 133 and line 137: How could equations (1) and (2) have the same right sides? Also, what is the time factor for the T here? Is it measured at hourly, daily, weekly or annually time period? Response: Thanks for your good suggestion. The equation (2) in the original manuscript was not correct. We have corrected equation (2). Here, the T represents the soil temperature recorded when measuring soil respiration. In this study, we only selected Q10 data when soil respiration measurement time was not less than four months (see section 2.1. Data collection). Here, the time period among case studies was not consistent with each other. Some studies provided the weekly time period, and some studies provided the monthly time period.

Line 155: Please define the "R-square and the model" here. Response: Thanks for your suggestion. We have defined the "R-square and the model" in the section 2.1. Data collection when the R2 first appeared in the manuscript.

Line 174: Why using "a constant of 0.58" here? I think it should be 0.5 now (see Pribyl 2010, Geoderma 156: 75–83). Response: Thanks for your good suggestion. We have carefully read the article you provided (Pribyl 2010), in which the author suggested that the constant of 0.50 is more accurate than the conventional factor of 0.58. At present, the conversion factor of 0.50 was widely used. We have converted soil organic matter to soil organic carbon by the constant of 0.50. Meanwhile, we re-analyzed the content related to soil organic carbon, and revised the corresponding text throughout the entire manuscript and the supplementary information.

Line 263: "Q10-ST10" is not shown by Figure 5. Did you mean Q10-ST5? Response: Thanks for your comment. Here, the Q10-ST5 was correct. In the original manuscript, the caption of Figure 5 missed the information of Q10-ST10, but the figures in Figure 5 were right. Now, we have added the missing information of Q10-ST10 in the caption of Figure 5.

Line 267: Not "Table S3", should be Table S4. Response: Thanks for your correction. We have changed Table S3 to Table S4.

Line 302: "untimely" should be 'ultimately' Response: Thanks for your correction. We have corrected the word.

Lines 308-315: The discussion here is unclear. Response: Thanks for your good suggestion. We have re-written this part of discussion (line 356-362).

Line 320: "n=20" here, but there were only 6 dots in the figure? Response: Thanks for your corrections. Here, we miswrote the sample size. Indeed, there only 6 dots for the relationships of Rs and belowground biomass. We have changed n = 20 to n = 6.

Lines 331-352: These low R-square values could be a serious problem for this manuscript. How did you deal with this issue? Response: Thank you for the comment. In this study, we obtained Q10 and its R2 calculated using the equation (1) and (2). We only selected the R2 values when the exponential fitting between soil respiration (Rs) and soil temperature were statistically significant (p < 0.05). If the p values were larger than 0.05 in case study, we did not select the Q10 and its R2 value. In spite of this, the R2 in some case studies were very low. As presented in this study, only 37.3% of R2 for Q10 was larger than 0.7, indicating that most of the seasonal variation of Rs rate cannot be well explained by soil temperature using the van't Hoff equation. In section 4.2.1 R2 for Q10 in Chinese grasslands, we discussed the R2 for Q10 in detail, and pointed out that for ecosystems (e.g., grassland and desert) in arid and semi-arid regions, Rs could be better estimated by the combined factors of soil temperature and moisture.

Lines 405-425: This section is really rough. The quality of the discussion needs improvement. Response: Thanks for your good suggestion. We have revised this part (line 470-500).

Lines 453-457: To me, Fig. 7 actually showed huge differences between those three methods. Response: Thanks for your comment. Here, we guess you mean Figure S7. The differences might be not only due to the measurement methods, but also be due to the differences among grassland types. To eliminate the influences of grassland

type, we also compared the measurement method effects within each grassland type. As presented in the new Figure S7, the ANOVA analyses showed that there were generally no significant differences for Q10 (at the soil depth of 5 and 10 cm) among measurement methods, whether the data was pooled across all grasslands or within each grassland type. For Rs, there was only one sample from alkali absorption (AA, Rs = 202.5), which seems to be much lower than dynamic closed chamber (DCC, Rs = 589.2) and static closed chamber (SCC, Rs = 459.9). Considering this AA data for Rs was from temperate typical steppe (TTS), we also compared this value (202.5) measured by AA to those measured by DCC and SCC within TTS. We found that the value of 202.5 (AA) was lower than 548.3 (DCC), but close to 193.0 (SCC). Therefore, including the single data measured by the alkali absorption method in our synthesis does not meaningfully change the results of Rs and Q10.

Lines 471-473: The sentence structure is problematic. Response: Thanks for your comment. We have re-written the sentence.

Lines 468-481: The Conclusion really needs lots of improvement. Response: Thanks for your good suggestion. We have revised this part.

―――――――――――――――――

---

## Author Comment (AC2) · 15 Jul 2018

General comments

In this paper, the authors used published data to analyze the variations of soil $CO_2$ respiration rates and their temperature sensitivity (Q10) across Chinese grasslands. Furthermore, their relationships with some abiotic and biotic factors were analyzed. The results could advance the understanding of the variation and control factors of soil $CO_2$ respiration rates and their temperature sensitivity (Q10).

Response: Thank you very much for your encouragement.

Specific comments:

Line 72: shown

Response: Thanks for your correction. We have corrected the word.

Line 137: Correct the equation 2

Response: Thanks for your good suggestion. We have corrected the equation. Please see line 154 in the text.

Line 148-151: The Q10 values were divided into five soil depth with different soil temperature

Response: Thanks for your good suggestion. We have revised the sentence. Please see line 172-173 in the text.

Line 178: shown

Response: Thanks for your correction. We have corrected the word.

Line S2: add the measuring methods

Response: Thanks for your good suggestion. We guess you mean add the measuring methods in line 187-188 in the original manuscript. Following your suggestion, we have added the measuring methods in the supplement file.

Line 192, Fig. 4: Why choose paired sample t-test to analyze the significant differences of the Q10 among the different soil depths?

Response: In this study, most studies reported the Q10 values derived by soil temperature at one or two different soil depths. For example, one study includes Q10 at 5 and 10 cm soil depth, one study includes Q10 at 10 and 15 cm soil depth, and another study includes Q10 at 10, 15 and 20 cm soil depth. Under this condition, the Q10 at the five soil depths was not paired. Therefore, when combining all Q10 from different studies and comparing Q10 derived by the five soil depths, the differences for Q10

among soil depths might be result from grassland type, rather than soil depth. Therefore, we choose paired sample t-test to analyze the significant differences of the Q10 among the different soil depths. When treating the similar data, previous studies also applied the paired sample t-test to analyze the significant differences, such as Peng et al (2009) and Wang et al (2010). However, several studies used one way analysis of variance (ANOVA) to compare the differences for Q10 among different soil depths, such as Song et al (2014), Xu et al (2015). Therefore, we also applied ANOVA to analyses the differences among different soil depths, which can also present the patterns of Q10 among soil depths. The results from paired sample t-test were presented in the manuscript (Fig. 4), and the results from ANOVA were presented in the supplement file (Fig. S3).

Line 209: there are no results for the temperate desert steppe in Table 1

Response: In this study, we focused on soil respiration at the annual scale. Meanwhile, we also checked the original data. Indeed, we found that there was no annual soil respiration measured in temperate desert steppe in China when we searched references. Therefore, our results for annual soil respiration rate did not include temperate desert steppe (Table 1), and we noted this condition in the captions of Table 1.

Line 233 and Line 239: five soil depths

Response: Thanks. We have corrected the writing.

Line 248: 1.73±0.08

Response: Thanks. We have changed 2.65±0.08 to 1.73±0.08.

Line 267: Table S4

Response: Thanks. We have corrected the writing.

Line 271-286, most of the contents are descriptive and repeated with results

Response: Thanks for your comment. We have re-written this part (line 305-324).

Line 364 relatively colder and higher than what?

Response: Thanks for your comment. We have described it in detail.

Fig. 2, 5: indicate the n values for each regression analysis

Response: Thanks for your good suggestion. We have indicated the n values for each regression analysis.

Fig. 3 Line 675 (e) and (g)

Response: Thanks. We have corrected the word.

Table S1: what $R^2$ represent for? What the ranges of soil temperature and soil moisture?

Response: Thanks. Here, in Table S1, the $R^2$ represent the determination coefficient for the relationship between soil temperature and soil respiration rate based on equation (1) and (2). In order to clearly distinguish this type of $R^2$ from the $R^2$ in regression analyses in Figure 2 and Figure 5, we changed all this type of $R^2$ to $RQ^2$ through the entire manuscript. We have revised the related descriptions in detail, please see the definition of $RQ^2$ in section 2.1 Data collection and related content in the revised manuscript. For soil temperature and soil moisture, these two parameters are provided with different time scale in case studies, for example some studies provided monthly or weekly mean temperature and moisture, some studies provided daily mean temperature and moisture, and some studies provided daily temperature and moisture. In this case, we could not accurately obtain the ranges of these two parameters, and we did not include the ranges of soil temperature and soil moisture in our dataset and analysis.

Table S2: show the n values. Are there values of soil temperature and soil moisture?

Response: Thanks. We have indicated the n values for each item in Table S2. Meanwhile, we have added soil temperature and soil moisture in Table S2. As the two key environmental factors, these two parameters might also control soil respiration and its

temperature sensitivity (Q10). Therefore, we also analyzed the relationships between these two parameters and annual soil respiration and Q10 derived by soil temperature at the depth of 5 and 10 cm, respectively.

Table S3: show the n values. Are there values of soil temperature and soil moisture?

Response: Thanks. We have indicated the n values in Table S3. Meanwhile, we have added analysis of soil temperature and soil moisture in Table S3.

Fig. S1, S5, S6: show the n values

Response: Thanks for your good suggestion. We have indicated the n values in Figure S1, S5 and S6.

Fig. S7: is data for method comparison from the same or similar sites? Otherwise, there may be many factors affect the annual Rs and Q10.

Response: Thanks for your comment. Here, the data for method comparison is from all sites. Indeed, when combining all data from different sites, the method comparisons for Rs and Q10 are affected by many factors, such as grassland types, soil properties. As presented in the section 3.2.1 and 3.2.3, the Rs and Q10 are affected by many environmental factors. Under this condition, one of the ways to address this issue is using data from the same or similar sites to compare the differences among measuring methods. We treated the grasslands within each grassland type as similar sites. Here, in order to eliminate the influences of other factors, we also compared the measurement method effects within each grassland type. As presented in the new Figure S7, the ANOVA analyses showed that there were generally no significant differences for Rs, Q10 derived by soil temperature at the depth of 5 and 10 cm among measurement methods, whether the data was pooled across all grasslands or within each grassland type. Due to the only one sample of annual Rs measured by alkali absorption (AA), we could not compare it to the other two methods using ANOVA analysis. Considering the value measured by AA was very close to that by static closed chamber (SCC), the

effects of measurement methods on Rs could be neglected.

References used in our responses:

Peng, S., Piao, S., Wang, T., Sun, J., and Shen, Z., 2009. Temperature sensitivity of soil respiration in different ecosystems in China. Soil Biology & Biochemistry, 41, 1008–1014.

Pribyl, D.W., 2010. A critical review of the conventional SOC to SOM conversion factor. Geoderma, 156(3–4), 75-83.

Song, X., Peng, C., Zhao, Z., Zhang, Z., Guo, B., Wang, W., Jiang, H., Zhu Q. 2014. Quantification of soil respiration in forest ecosystems across china. Atmospheric Environment, 94, 546–551.

Wang, W., Chen, W., and Wang, S. 2010. Forest soil respiration and its heterotrophic and autotrophic components: Global patterns and responses to temperature and precipitation. Soil Biology & Biochemistry, 42, 1236–1244.

Wang, X., Piao, S., Ciais, P., Janssens, I.A., Reichstein, M., Peng, S., and Wang, T. 2010. Are ecological gradients in seasonal Q10 of soil respiration explained by climate or by vegetation seasonality? Soil Biology & Biochemistry, 42, 1728–1734.

Xu, Z., Tang, S., Xiong, L., Yang, W., Yin, H., Tu, L., Wu, F., Chen, L., and Tan, B. 2015. Temperature sensitivity of soil respiration in china's forest ecosystems: patterns and controls. Applied Soil Ecology, 93, 105–110.

––––––––––––––––––––––––––––––––

---

## Author Response (AR1)

**Associate Editor Decision: Publish subject to minor revisions (review by editor) (05 Aug 2018) by Zhongjun Jia**

Comments to the Author:

Dear Dr. Zhu,

Thank you for submitting your revised MS to BG

I had a quick look at your manuscript, and feel that the major concerns have been adequately addressed.

However, I would like to raise your concern about some minor points.

(1) The title can be rephrased as: Patterns of soil respiration and its temperature sensitivity in grassland ecosystems across China. As you can see, both reviewers, particularly reviewer#1 has major concern about the volume of your sample size. Although your reply appears reasonable, the key driver of soil respiration and its temperature sensitivity is not conclusively deciphered. I would like to say no single paper could resolve this problem, ant your attempt to tackle this question is welcomed. In addition, the need for more measurements might be highlighted at the text in the conclusion section.

Response: Thanks for your good suggestions. We have changed our original title to "*Patterns of soil respiration and its temperature sensitivity in grassland ecosystems across China*". In addition, we have highlighted the need for more measurements for soil respiration and its temperature sensitivity. Please see line 585–591 in the revised manuscript ("track change" version).

(2) The advantage and disadvantage of ANOVA and paired *t* text need to be discussed for its ecological implication, rather than simply stating the methods used.

Response: Thanks for your good suggestions. In our manuscript, we mainly used ANOVA and paired *t* to explore the differences among groups. Here, the paired *t* test was used to compare the differences between growing season and non-growing season soil respiration (Rs), and between autotrophic respiration and heterotrophic respiration, and the $Q_{10}$ values among different measurement depths from same sites, because these variables were from the same sites and in one-to-one correspondence. In addition, in our manuscript, we used both ANOVA and paired $t$ to examine the effects of depth on $Q_{10}$. To clarify the ecological implication between these two statistical methods, we have briefly described their differences. Please see line 220–228 in the revised manuscript ("track change" version).

Yours sincerely

Zhongjun

**Responses to reviewers' comments on the manuscript bg-2018-83**

**Title:** Patterns and controls of soil respiration and its temperature sensitivity in grassland ecosystems across China

**Authors:** Jiguang Feng, Jingsheng Wang, Yanjun Song, Biao Zhu

Dear Dr. Jia,

Thank you very much for your kind work. Both reviewers' comments are very constructive and helpful. We have considered these comments and made a major revision of original manuscript. In the revised manuscript, we used "track changes" option to highlight where we revised, and we show the detailed response (in blue text) to each comment in this document.

We are looking forward to receiving your decision.

Thank you and best regards.

Yours sincerely,

Dr. **Biao Zhu** (Corresponding author) on behalf of all authors

College of Urban and Environmental Sciences, Peking University, Beijing 100871, China
Telephone: +86 10 62745258
Email address: biaozhu@pku.edu.cn

**Referees' comments:**

**Anonymous Referee #1:**

*General comments*

This manuscript made a contribution by compiling published data of soil respiration and temperature sensitivity related to soil respiration from five types of Chinese grasslands. The spatial extend of the dataset covers a large region. The temporal extend of the dataset is at the annual scale. It seems that the majority of the data points in this dataset have not been integrated into any published synthesis yet. Some aspects of the manuscript deserve attention. The authors carried out some basic correlation analyses on this dataset, and found some inconsistencies as compared with results in some published reports. One inconsistency was the correlation between annual soil respiration rate (Rs) and total soil nitrogen content (or total soil carbon content, because soil C and N tend to go together). As normally expected, most published reports showed highly significant correlation between Rs and soil C & N, but not this manuscript. The actual causes of this inconsistency were unclear. Another inconsistency was that the manuscript did not find any significant correlations between climatic variable (e.g., temperature and precipitation) and $Q_{10}$ values measured at 5 cm or 10 cm depth, which is in contrast to published results. Again, clear causes of this inconsistency were not offered.

Response: Thanks for the constructive comments. We show our response to the three main comments on the inconsistency between our results and previous studies.

The first inconsistency was the correlation between annual soil respiration rate (Rs) and total soil nitrogen. In this study, we found that annual soil respiration did not significantly correlate with soil total nitrogen ($p = 0.10$, Fig. 2f), which was not consistent with previous results at the regional and global scale. Not surprisingly, we found that soil organic carbon was closely associated with soil total nitrogen ($p < 0.01$, Table S3). But, annual soil respiration increased closely with soil organic carbon ($p < 0.001$, Fig. 2e). The non-significant correlation between soil total nitrogen and annual soil respiration might be due to the limited sample size in soil total nitrogen compared to soil organic carbon (24 vs. 40), and/or due to the fact that soil total nitrogen might not well represent nitrogen availability for plants and microbes.

The second inconsistency was that this study did not find any significant correlations between climatic variables (i.e. mean annul temperature (MAT) and mean annual precipitation (MAP)) and $Q_{10}$ values measured at 5 or 10 cm depth. This was not consistent with previously published results. But, we found $Q_{10}$ measured at 5 or 10 cm soil depth was significantly decreased with increasing soil temperature, partly supporting the previous statement that $Q_{10}$ tends to be higher in colder regions. Additionally, although the single factor of precipitation or temperature only explained a small proportion of the spatial variation of $Q_{10}$, the combined factors of MAT and MAP, or soil temperature and soil moisture, explained a significant proportion of the spatial variation of $Q_{10}$ across Chinese grasslands at regional scale (Table S4). Please see the discussion in section *4.2.3 Controls of environmental factors on $Q_{10}$.*

As the authors stated in the manuscript, the soil respiration in this context has two main components: autotrophic respiration of plant roots, and heterotrophic respiration of soil microbes. Therefore, the soil respiration should be controlled by both plant-related variables and soil-related variables. But unfortunately, there were only 7 data points that have autotrophic and heterotrophic respiration measured separately (and probably using questionable methods). Consequently, Rs and $Q_{10}$ data could not be discussed in relations to plant-related variables and soil-related variables. Furthermore, these $Q_{10}$ values were calculated using the seasonally changing temperature data which often highly co-vary with plant growth (therefore, the seasonal increase of root respiration). As a result, the seasonal increase of root respiration would contribute to abnormally high $Q_{10}$ values. This key aspect definitely needs authors' attention. Changes in the Introduction, Materials and Methods, and Discussion sections are required accordingly.

Response: Thanks for the constructive comments. When discussing annual soil respiration among grassland types, we analyzed autotrophic (root) respiration and heterotrophic (microbial) respiration, respectively, for example, section *4.1.1 Annual soil*

*respiration among grassland types*. In addition, as the substrate of microbial decomposition, soil organic carbon (SOC) affects soil respiration. In addition, soil pH mainly controls heterotrophic respiration via regulating soil microbial activities. Therefore, the discussions related to SOC and pH were associated with plant-related variables and soil-related variables. But, the few samples ($n = 7$) from heterotrophic respiration and autotrophic respiration measured separately limited the in-depth discussions. We have pointed out this issue and highlighted the needs for more measurements in the section *5 Conclusion*.

As you stated, the seasonal dynamics of plant growth affect root respiration and thereby seasonal $Q_{10}$. At large scale, the seasonal amplitude of plant activity among different sites varied largely, which could affect the calculated seasonal $Q_{10}$. Indeed, a previous global synthesis study found that seasonal amplitude of plant activity fundamentally dominates seasonal $Q_{10}$ among different study sites compared with other environmental factors (Wang et al. 2010). But, in this study, we could not analyze the effects of seasonal variation of root respiration on $Q_{10}$ due to the limited samples ($n = 7$) from autotrophic (root) respiration. In addition, the seasonal dynamics of plant growth at a given site might also affect the calculated $Q_{10}$. In this study, our dataset included $Q_{10}$ estimated at different time scale for measuring soil respiration. We categorized them into three types according to plant growth stage, including growing season $Q_{10}$, non-growing season $Q_{10}$, and annual $Q_{10}$. In this case, we also conducted a one-way ANOVA analysis to examine the effects of measurement period (including growing season, non-growing season and annual scale) on $Q_{10}$ derived by soil temperature at the depth of 5 and 10 cm, and found that measurement period did not significantly affect $Q_{10}$ derived by soil temperature at the depth of 10 cm, but significantly affected $Q_{10}$ derived by soil temperature at the depth of 5 cm. (Fig. S7). We have discussed this result in section *4.4 Uncertainties*. Following your suggestions, we have revised the related content in the sections of Introduction, Materials and Methods (Section *2.2 Data analysis*), and Discussion (section *4.2.3 Controls of environmental factors on Q10* and Section *4.4 Uncertainties*).

*Specific/Minor comments*

Line 25: 'latitude and' should be removed here. These geographic features (e.g., latitude, longitude, altitude or elevation) may be used as proxies for temperature or precipitation in data analysis only when temperature or precipitation data were not available. So authors should consider eliminate all parts of the manuscript that use these geographic features in statistical analyses and any related discussion.

Response: Thanks for your good suggestion! We have eliminated all parts of the manuscript that use these geographic features in statistical analyses and any related discussion. In the revised manuscript, we added statistical analyses and discussion of soil temperature and soil moisture.

Line 28: The % heterotrophic respiration was only based on 7 data points, therefore, should not be in the abstract. Similarly, if the authors really want to make the "key" point of growing season vs. non-growing season, they should have given clear descriptions about how the separation was done accurately and reliably.

Response: Thanks for your good suggestions! We have eliminated the contents related to % heterotrophic respiration and % growing season in the abstract. In addition, we described how the growing season and non-growing season were defined. The growing season was from May to October, and the non-growing season was from November to April in the second year.

Lines 29-31: This sentence needs a re-write so that the meaning becomes clear.

Response: Thanks for your good suggestion. We have re-written the sentence in the revised manuscript.

Line 33: Remove the sentence about latitude and longitude here (the reason is given at line 25).

Response: Thanks for your good suggestions. We have removed all the sentences related to latitude and longitude in the revised manuscript.

Lines 35-38: Authors need to substantiate about 'how have they advanced the understanding' here.

Response: Thanks for your good suggestion. We have revised the abstract and substantiated which understandings were advanced (line 41-45).

Line 53: "on the large scale"? Do you really want to 'step' on the large scale by the wall? My guess is that you really want to state: 'at a large scale' here. This correction should be made throughout the entire manuscript.

Response: Thanks for your good suggestions. We have changed "on the large scale" to "at a large scale" throughout the entire manuscript.

Lines 67-68: Move the "and" to the place before the last part of the sentence, before "leaf area index".

Response: Thank you. We have moved the "and" before "leaf area index".

Line 83: "As known to all …" The sentence is awkward.

Response: Thank you. We have re-written the sentence.

Line 133 and line 137: How could equations (1) and (2) have the same right sides? Also, what is the time factor for the $T$ here? Is it measured at hourly, daily, weekly or annually time period?

Response: Thanks for your good suggestion. The equation (2) in the original manuscript was not correct. We have corrected equation (2). Here, the T represents the soil temperature recorded when measuring soil respiration. In this study, we only selected $Q_{10}$ data when soil respiration measurement time was not less than four months (see section *2.1. Data collection*). Here, the time period among case studies was not consistent with each other. Some studies provided the weekly time period, and some studies provided the monthly time period.

Line 155: Please define the "R-square and the model" here.

Response: Thanks for your suggestion. We have defined the "R-square and the model" in the section *2.1. Data collection* when the $R^2$ first appeared in the manuscript.

Line 174: Why using "a constant of 0.58" here? I think it should be 0.5 now (see Pribyl 2010, Geoderma 156: 75–83).

Response: Thanks for your good suggestion. We have carefully read the article you provided (Pribyl 2010), in which the author suggested that the constant of 0.50 is more accurate than the conventional factor of 0.58. At present, the conversion factor of 0.50 was widely used. We have converted soil organic matter to soil organic carbon by the constant of 0.50. Meanwhile, we re-analyzed the content related to soil organic carbon, and revised the corresponding text throughout the entire manuscript and the supplementary information.

Line 263: "$Q_{10\text{-ST10}}$" is not shown by Figure 5. Did you mean $Q_{10\text{-ST5}}$?

Response: Thanks for your comment. Here, the $Q_{10\text{-ST5}}$ was correct. In the original manuscript, the caption of Figure 5 missed the information of $Q_{10\text{-ST10}}$, but the figures in Figure 5 were right. Now, we have added the missing information of $Q_{10\text{-ST10}}$ in the caption of Figure 5.

Line 267: Not "Table S3", should be Table S4.

Response: Thanks for your correction. We have changed Table S3 to Table S4.

Line 302: "untimely" should be 'ultimately'

Response: Thanks for your correction. We have corrected the word.

Lines 308-315: The discussion here is unclear.

Response: Thanks for your good suggestion. We have re-written this part of discussion (line 364-370).

Line 320: "n=20" here, but there were only 6 dots in the figure?

Response: Thanks for your corrections. Here, we miswrote the sample size. Indeed, there only 6 dots for the relationships of Rs and belowground biomass. We have changed $n = 20$ to $n = 6$.

Lines 331-352: These low R-square values could be a serious problem for this manuscript. How did you deal with this issue?

Response: Thank you for the comment. In this study, we obtained $Q_{10}$ and its $R^2$ calculated using the equation (1) and (2). We only selected the $R^2$ values when the exponential fitting between soil respiration (Rs) and soil temperature were statistically significant ($p < 0.05$). If the $p$ values were larger than 0.05 in case study, we did not select the $Q_{10}$ and its $R^2$ value. In spite of this, the $R^2$ in some case studies were very low. As presented in this study, only 37.3% of $R^2$ for $Q_{10}$ was larger than 0.7, indicating that most of the seasonal variation of Rs rate cannot be well explained by soil temperature using the van't Hoff equation. In section *4.2.1 $R^2$ for $Q_{10}$ in Chinese grasslands*, we discussed the $R^2$ for $Q_{10}$ in detail, and pointed out that for ecosystems (e.g., grassland and desert) in arid and semi-arid regions, Rs could be better estimated by the combined factors of soil temperature and moisture.

Lines 405-425: This section is really rough. The quality of the discussion needs improvement.

Response: Thanks for your good suggestion. We have revised this part (line 478-508).

Lines 453-457: To me, Fig. 7 actually showed huge differences between those three methods.

Response: Thanks for your comment. Here, we guess you mean Figure S7. The differences might be not only due to the measurement methods, but also be due to the differences among grassland types. To eliminate the influences of grassland type, we also compared the measurement method effects within each grassland type. As presented in the new Figure S7, the ANOVA analyses showed that there were generally no significant differences for $Q_{10}$ (at the soil depth of 5 and 10 cm) among measurement methods, whether the data was pooled across all grasslands or within each grassland type. For Rs, there was only one sample from alkali absorption (AA, Rs = 202.5), which seems to be much lower than dynamic closed chamber (DCC, Rs = 589.2) and static closed chamber (SCC, Rs = 459.9). Considering this AA data for Rs was from temperate typical steppe (TTS), we also compared this value (202.5) measured by AA to those measured by DCC and SCC within TTS. We found that the value of 202.5 (AA) was lower than 548.3 (DCC), but close to 193.0 (SCC). Therefore, including the single data measured by the alkali absorption method in our synthesis does not meaningfully change the results of Rs and $Q_{10}$.

Lines 471-473: The sentence structure is problematic.

Response: Thanks for your comment. We have re-written the sentence.

Lines 468-481: The Conclusion really needs lots of improvement.

Response: Thanks for your good suggestion. We have revised this part.

**Anonymous Referee #2:**

*General comments*

In this paper, the authors used published data to analyze the variations of soil $CO_2$ respiration rates and their temperature sensitivity ($Q_{10}$) across Chinese grasslands. Furthermore, their relationships with some abiotic and biotic factors were analyzed. The results could advance the understanding of the variation and control factors of soil $CO_2$ respiration rates and their temperature sensitivity ($Q_{10}$).

Response: Thank you very much for your encouragement.

*Specific comments:*

Line 72: shown

Response: Thanks for your correction. We have corrected the word.

Line 137: Correct the equation 2

Response: Thanks for your good suggestion. We have corrected the equation. Please see line 154 in the text.

Line 148-151: The $Q_{10}$ values were divided into five soil depth with different soil temperature

Response: Thanks for your good suggestion. We have revised the sentence. Please see line 172-173 in the text.

Line 178: shown

Response: Thanks for your correction. We have corrected the word.

Line S2: add the measuring methods

Response: Thanks for your good suggestion. We guess you mean add the measuring methods in line 187-188 in the original manuscript. Following your suggestion, we have added the measuring methods in the supplement file.

Line 192, Fig. 4: Why choose paired sample $t$-test to analyze the significant differences of the $Q_{10}$ among the different soil depths?

Response: In this study, most studies reported the $Q_{10}$ values derived by soil temperature at one or two different soil depths. For example, one study includes $Q_{10}$ at 5 and 10 cm soil depth, one study includes $Q_{10}$ at 10 and 15 cm soil depth, and another study includes $Q_{10}$ at 10, 15 and 20 cm soil depth. Under this condition, the $Q_{10}$ at the five soil depths was not paired. Therefore, when combining all $Q_{10}$ from different studies and comparing $Q_{10}$ derived by the five soil depths, the differences for $Q_{10}$ among soil depths might be result from grassland type, rather than soil depth. Therefore, we choose paired sample $t$-test to analyze the significant differences of the $Q_{10}$ among the different soil depths. When treating the similar data, previous studies also applied the paired sample $t$-test to analyze the significant differences, such as Peng et al (2009) and Wang et al (2010). However, several studies used one way analysis of variance (ANOVA) to compare the differences for $Q_{10}$ among different soil depths, such as Song et al (2014), Xu et al (2015). Therefore, we also applied ANOVA to analyses the differences among different soil depths, which can also present the patterns of $Q_{10}$ among soil depths. To clarify the ecological implication between these two statistical methods, we have briefly described their differences. Please see line 220–228 in the revised manuscript. The results from paired sample *t*-test were presented in the manuscript (Fig. 4), and the results from ANOVA were presented in the supplement file (Fig. S3).

Line 209: there are no results for the temperate desert steppe in Table 1

Response: In this study, we focused on soil respiration at the annual scale. Meanwhile, we also checked the original data. Indeed, we found that there was no annual soil respiration measured in temperate desert steppe in China when we searched references. Therefore, our results for annual soil respiration rate did not include temperate desert steppe (Table 1), and we noted this condition in the captions of Table 1.

Line 233 and Line 239: five soil depths

Response: Thanks. We have corrected the writing.

Line 248: 1.73±0.08

Response: Thanks. We have changed 2.65±0.08 to 1.73±0.08.

Line 267: Table S4

Response: Thanks. We have corrected the writing.

Line 271-286, most of the contents are descriptive and repeated with results

Response: Thanks for your comment. We have re-written this part (line 313-332).

Line 364 relatively colder and higher than what?

Response: Thanks for your comment. We have described it in detail.

Fig. 2, 5: indicate the n values for each regression analysis

Response: Thanks for your good suggestion. We have indicated the *n* values for each regression analysis.

Fig. 3 Line 675 (e) and (g)

Response: Thanks. We have corrected the word.

Table S1: what $R^2$ represent for? What the ranges of soil temperature and soil moisture?

Response: Thanks. Here, in Table S1, the $R^2$ represent the determination coefficient for the relationship between soil temperature and soil respiration rate based on equation (1) and (2). In order to clearly distinguish this type of $R^2$ from the $R^2$ in regression analyses in Figure 2 and Figure 5, we changed all this type of $R^2$ to $R_Q^2$ through the entire manuscript. We have revised the related descriptions in detail, please see the definition of $R_Q^2$ in section *2.1 Data collection and related content in the revised manuscript*. For soil temperature and soil moisture, these two parameters are provided with different time scale in case studies, for example some studies provided monthly or weekly mean temperature and moisture, some studies provided daily mean temperature and moisture, and some studies provided daily temperature and moisture. In this case, we could not accurately obtain the ranges of these two parameters, and we did not include the ranges of soil temperature and soil moisture in our dataset and analysis.

Table S2: show the *n* values. Are there values of soil temperature and soil moisture?

Response: Thanks. We have indicated the *n* values for each item in Table S2. Meanwhile, we have added soil temperature and soil moisture in Table S2. As the two key environmental factors, these two parameters might also control soil respiration and its temperature sensitivity ($Q_{10}$). Therefore, we also analyzed the relationships between these two parameters and annual soil respiration and $Q_{10}$ derived by soil temperature at the depth of 5 and 10 cm, respectively.

Table S3: show the *n* values. Are there values of soil temperature and soil moisture?

Response: Thanks. We have indicated the *n* values in Table S3. Meanwhile, we have added analysis of soil temperature and soil moisture in Table S3.

Fig. S1, S5, S6: show the *n* values

Response: Thanks for your good suggestion. We have indicated the n values in Figure S1, S5 and S6.

Fig. S7: is data for method comparison from the same or similar sites? Otherwise, there may be many factors affect the annual Rs and $Q_{10}$.

Response: Thanks for your comment. Here, the data for method comparison is from all sites. Indeed, when combining all data from different sites, the method comparisons for Rs and $Q_{10}$ are affected by many factors, such as grassland types, soil properties. As presented in the section 3.2.1 and 3.2.3, the Rs and $Q_{10}$ are affected by many environmental factors. Under this condition, one of the ways to address this issue is using data from the same or similar sites to compare the differences among measuring methods. We treated the grasslands within each grassland type as similar sites. Here, in order to eliminate the influences of other factors, we also compared the measurement method effects within each grassland type. As presented in the new Figure S7, the ANOVA analyses showed that there were generally no significant differences for Rs, $Q_{10}$ derived by soil temperature at the depth of 5 and 10 cm among measurement methods, whether the data was pooled across all grasslands or within each grassland type. Due to the only one sample of annual Rs measured by alkali absorption (AA), we could not compare it to the other two methods using ANOVA analysis. Considering the value measured by AA was very close to that by static closed chamber (SCC), the effects of measurement methods on Rs could be neglected.

**References used in our responses:**

Peng, S., Piao, S., Wang, T., Sun, J., and Shen, Z., 2009. Temperature sensitivity of soil respiration in different ecosystems in China. *Soil Biology & Biochemistry*, 41, 1008–1014.

Pribyl, D.W., 2010. A critical review of the conventional SOC to SOM conversion factor. Geoderma, 156(3–4), 75-83.

Song, X., Peng, C., Zhao, Z., Zhang, Z., Guo, B., Wang, W., Jiang, H., Zhu Q. 2014. Quantification of soil respiration in forest ecosystems across china. *Atmospheric Environment*, 94, 546–551.

Wang, W., Chen, W., and Wang, S. 2010. Forest soil respiration and its heterotrophic and autotrophic components: Global patterns and responses to temperature and precipitation. *Soil Biology & Biochemistry*, 42, 1236–1244.

[revised manuscript text omitted]

---

## Author Response (AR2)

**Response to the Associate Editor Decision: Publish subject to technical corrections (bg-2018-83)**

Dear Prof. Jia,

Thank you very much for the corrections. We have revised our manuscript according to your comments. We are looking forward to receiving your decision.

Thank you and best regards.

Yours sincerely,

Dr. **Biao Zhu (corresponding author) on behalf of all authors**

**Associate Editor Decision: Publish subject to technical corrections (22 Aug 2018) by Zhongjun Jia**

Comments to the Author:

Dear Prof. Zhu

Thank you for submitting the revised ms to BGD.

The comments and concerns have been adequately addressed. The manuscript can be published and the following technical concerns for your reference.

(1) L588. delete strongly

Response: We have deleted the word.

(2) L590 delete fundamentally

Response: We have deleted the word.

Once again, thanks for your submission to BG

Kind regards

Zhongjun Jia